# Evaluation of Biostimulatory Activity of Commercial Formulations on Three Varieties of Chickpea

Regina Gabilondo [1], Jorge Sánchez [1], Pedro Muñoz [1], Iris Montero-Muñoz [1,†], Pedro V. Mauri [1], José Marín [2] and David Mostaza-Colado [1,*]

1 Instituto Madrileño de Investigación y Desarrollo Rural, Agrario y Alimentario (IMIDRA) Finca El Encín, Autovía A-2. Km. 38,200, 28805 Alcalá de Henares, Spain
2 Área Verde MG Projects S.L., C/Oña 43, Bajo, 28050 Madrid, Spain
* Correspondence: david.mostaza@madrid.org
† Current address: Real Jardín Botánico (RJB), CSIC, 28014 Madrid, Spain.

**Abstract:** Biostimulants are studied as a possible agricultural practice that anticipates the reproductive stages of chickpeas to avoid their coincidence with high temperatures and hydric stress periods. The effect of several types of biostimulants on different chickpea varieties was analyzed. The Blanco Sinaloa chickpea variety showed opposite patterns with respect to biostimulant effect on germination success and vegetative and radicular development when compared with two other chickpea varieties, namely Amelia, a well-known variety, and IMIDRA10, a recently developed variety. Blanco Sinaloa is cultured under water irrigation conditions, while Amelia and IMIDRA10 are used under rainfed conditions. Blanco Sinaloa and IMIDRA10 are Kabuli-type varieties, while Amelia is Desi-type. All varieties emerged 9 days after the sowing, but Amelia nascence was more abundant at the beginning, on day 9. On day 32, the picture was quite different, since Blanco Sinaloa had germinated 100% in practically all treatments, followed by Amelia and IMIDRA10. There were significant differences between plant lengths among the three varieties, since Blanco Sinaloa is much larger than Amelia and IMIDRA10. Blanco Sinaloa was the only variety in which the plant lengths of biostimulant-impregnated seeds were superior to those of untreated plants; that is, it was the only one that was positively affected by biostimulants. Chickpea seeds should be treated with biostimulants such that they are dry for sowing, because the mechanic seeder only works with dry seeds.

**Keywords:** *Cicer arietinum* L.; Blanco Sinaloa; Amelia; PGPR; plant growth-promoting rhizobacteria; *Trichoderma*; *Bacillus*; *Glomus*; endomycorrhiza





## 1. Introduction

To improve the efficiency of the chickpea (*Cicer arietinum* L.) crop and adapt it to climate change, the effects of biostimulants on germination success and vegetative development have been studied. Because of climate change, the average annual temperature has increased while rainfall has decreased, especially when the plant develops into its reproductive stages. This harms chickpea yields in both conventional and ecological management [1]. As an adaptation to climate change, agricultural practices that anticipate the reproductive stages of chickpea plants to avoid their coincidence with periods of high temperatures and hydric stress should be developed, i.e., the development of chickpea varieties with an earlier date of sowing or the use of biostimulants that accelerate and improve the vegetative development of the plants.

Regulation (EU) 2019/1009 [2] defines the term biostimulant as "an EU fertilizer product whose function is to stimulate the nutritional processes of plants regardless of the nutrient content of the product, with the sole objective of improving one or more of the following plant characteristics and its rhizosphere: efficiency in the use of nutrients, tolerance to abiotic stress, quality characteristics or availability of nutrients immobilized in

the soil and the rhizosphere". There are different types of biostimulants [3]: those based on beneficial microorganisms (bacteria and fungi), algae, and chitosan products. In this research work, different mixtures of beneficial bacteria and fungi, algae, plant extracts, and organic fertilizer were used based on the commercial formulations and the experience of farmers. The group called "Plant Growth-Promoting Rhizobacteria" (PGPR) is one of the most used treatments in chickpea plants [4] and was analyzed in the present work. Rhizobacteria are atmospheric nitrogen-fixing microorganisms that form mycorrhizae with the chickpea root and incorporate nitrogen into the plant and soil [5]. Plant growth and formation of mycorrhizae nodules in chickpea after applying biostimulants that contain beneficial microorganisms have been studied by other research groups and found to have positive effects [3,6–10]. PGPR are also involved in the production and release of growth phytohormones such as giberelines and cytokines [3]. The fungi *Trichoderma*, used in this experiment, improves the immune system of the plant, making it resistant to attacks by other fungi [11–13]. The fungus *Glomus intraradices,* also used herein, forms endomycorrhiza, which incorporates nitrogen and other nutrients into the plant [14]. Other variables have been analyzed in chickpea cultivations in which biostimulants had been applied with beneficial microorganisms: diameter, total plant mass, dry mass of nodules, leaf area index, greenness index, volatile compounds, yield (number of pods, pod mass, number of grains, mass of grains, grain protein content, biofortification), sprout and root length, N, P, and K uptake, the incidence of several diseases and pests, etc. [2]. All these variables showed positive effects of biostimulants on chickpea cultivation [2], i.e., higher protein content in the grain [7,15]; biofortification [15]; higher absorption of N, P, and K; increased activity of the enzymes SOD (superoxide dismutase) and POD (peroxidase); and an increase in the concentrations of organic acids, thus reducing the pH of the rhizosphere [16]. Infections by phytopathogens were also analyzed, and the results showed that they were inhibited or reduced [17–19]. Moreover, biostimulants based on PGPB respect the environment [20,21], and the degradation and contamination of the soil produced by agrochemicals is avoided. They contribute to the restoration of the soil microbial balance, and even abiotic stress is reduced [3,22].

The purpose of the present work is the study of the germination success (percentage and date after the sowing) and vegetative development (plant length and number of nodes) of three different varieties of chickpea, Amelia, IMIDRA10, recently developed, and Blanco Sinaloa, when different biostimulants were applied in liquid form onto the seeds 24 h before sowing. Amelia and IMIDRA10 are chickpea varieties used in rainfed conditions, while Blanco Sinaloa is a chickpea variety grown with water irrigation.

Another purpose of the present research study was to contribute to the development of a protocol of biostimulant application to chickpea plants that is useful for all types of agricultural practices, including intensive methods, taking into account that seeds should be dry for the sowing, as the mechanic seeder only works with dry seeds. The application of the biostimulants occurred prior to sowing. Seeds were impregnated with biostimulants 24 h before sowing in order to allow them to dry.

The proposed research hypothesis is the existence of a positive influence of the different biostimulants on chickpea plants in terms of germination and vegetative development. The null hypothesis is the absence of biostimulant influence on chickpea germination and vegetative development.

## 2. Materials and Methods

The effects of several types of biostimulants were analyzed on three different varieties of chickpea grown in flowerpots in a greenhouse of El Encín-IMIDRA [23], located in the municipality of Alcalá de Henares, Madrid, Spain, 40°31″17″ N, 3°17′27″ O. The temperature range in the greenhouse was 3.9–31.9 °C, with an average temperature $12.8 \pm 4.5$ °C. The light intensity range was 0.0–79,911.6 luxes, with an average light intensity of $6543.5 \pm 13,047.1$ luxes.

## 2.1. Chickpea Varieties Investigated

The names of the chickpea varieties used for the experiment are Amelia, IMIDRA10, and Blanco Sinaloa. Amelia is a Desi-type variety that has been improved through different research projects in Instituto Madrileño de Investigación y Desarrollo Rural, Agrario y Alimentario (IMIDRA). It is highly adapted to different terrains, especially dry ones, and resistant to important fungal diseases. It is well-known and marketed in the agricultural sector, it has high performance (kg/ha), and is grown under rainfed conditions. IMIDRA10 is a Kabuli-type variety, also improved in IMIDRA for high productivity and resistance to plagues. It is outstanding because of its high protein digestibility index (90.56%) and its low content of the antinutrient phytic acid. These digestibility characteristics, good taste, appearance, and texture make this chickpea variety ideal for Madrid stew. It is also grown under rainfed conditions. Finally, Blanco Sinaloa is another Kabuli-type variety cultured in America, northwest of Mexico, under water irrigation conditions. It is outstanding because of its very large size and performance (kg/ha) [24].

## 2.2. Biostimulant Formulations

Different types of biostimulants were analyzed, and the protocol of seed impregnation is described. The proportion of biostimulant per seed followed the indications on the label by the manufacturer.

PGPR + *Trichoderma*: This biostimulant mixture was composed of the following species of Plant Growth-Promoting Rhizobacteria (PGPR): *Bacillus subtilis*, *B. polymyxa*, *B. megaterium,* and *Pseudomonas fluorescens*; some species of fungi *Trichoderma* (*T. harzianum*, *T. reesei*, *T. viride*, *Gliocladium virens*); unspecified plant origin proteins that provided amino acids; *Yucca schidigera* extract; and the marine algae *Ascophyllum nodosum* extract. The mixture proportion consisted of 16.6 g of product and 100 mL distilled water. First, 40 mL of solution (6.64 g of product) was prepared, and 100 g of seeds were impregnated with 1 mL of the solution. PGPR + EM: The following PGPR species were present in the biostimulant mixture: *Azospirillum brasilense*, *Azotobacter chroococcum*, *Bacillus megaterium*, and *Pseudomonas fluorescens*; the fungus species *Glomus intraradices*, which forms endomycorrhiza (EM); the plant vitamins biotine, folic acid, B, B2, B3, B6, B7, B12, C, and K; unspecified protein hydrolysate; *Yucca schidigera* extract; and *Ascophyllum nodosum* extract. The mixture proportion consisted of: 16.6 g of product and 100 mL distilled water. First, 40 mL of solution (6.64 g of product) was prepared and 100 g of seeds were impregnated with 1 mL of the solution. OF: Nitrogen and potassium (NK) liquid organic fertilizer (OF) had a vegetal origin. Its declared contents were as follows: total nitrogen (N) 2.5%; organic nitrogen (N) 2.0%; potassium soluble in water ($K_2O$) 5.0%; total organic carbon (C) 23.0%; C/N 11.0%; total organic matter 35.0%; pH = 5–6; and density = 1.22–1.24 gr/$cm^3$. The impregnation proportion was 1 mL of OF/100 g seeds. *Bacillus* sp. refers to *Bacillus paralicheniformis* water-soluble concentrate in proportion 0.25 mL of product/100 g seeds. 1 mL of product and 3 mL of distilled water were mixed, and 100 g of seeds were impregnated with 1 mL of the solution. PGPR + *Trichoderma* + OF: The mixture proportion was 16.6 g of PGPR + *Trichoderma* product/100 mL organic fertilizer. First, 40 mL of solution was prepared and 100 g of seeds were impregnated with 1 mL of the solution. Water was substituted by the organic fertilizer.

PGPR + EM + OF: the mixture proportion was 16.6 g of PGPR + EM product/100 mL organic fertilizer. First, 40 mL of solution were prepared and 100 g of seeds were impregnated with 1 mL of the solution. Water was substituted by the organic fertilizer.

## 2.3. Protocol of Seed Impregnation with Biostimulants

The chickpea seeds were impregnated 24 h before sowing, and were allowed to dry for 24 h so that they would be dry at the time of sowing. Different mixtures of biostimulants were prepared, 1 mL of the solution was applied in 100 g of chickpeas. A dry, untreated control sample was also prepared, as well as two controls in which 100 g of seeds were impregnated with either 1 mL or 10 mL of distilled water, in order to verify whether

the humidification of the seeds prior to sowing affected the germination or vegetative development. After soaking the 100 g samples of chickpea grains in 1 mL of the liquid biostimulant, they were shaken 20 times. Then, they were transferred to Petri dishes to dry for 24 h so they would be planted already dry.

### 2.4. Greenhouse Assay

Once in the greenhouse, a series of 11 cm × 11 cm and 14 cm deep pots was placed with a substrate mixed with 10% perlite. An initial abundant irrigation was carried out one hour before sowing, on 15 November 2022. Subsequently, one chickpea seed per pot was placed one centimeter deep and covered with the substrate and perlite. Four repetitions of each treatment were performed. Although chickpea is a rainfed crop, given the growing conditions in pots and greenhouses, watering was necessary, so 55 mL of water was added periodically. The pots were protected with mesh to avoid damage by birds or other animals.

### 2.5. Variables Studied

The parameters which we evaluated were germination success (percentage of germinated seeds in the assay), vegetative development (plant length and number of nodes), and phenological evolution of the crop, following the Schwartz and Langham scale [25]. Data collection of these variables was carried out every 3–4 days using a metric ruler. As the chickpeas were grown in a greenhouse, only the vegetative development (length and number of nodes) was studied; the yields could not be studied.

On day 42, the last day of the culture, plants were extracted from the pots. Roots were analyzed in search of root nodules, and in one of the replicas, the number of ramifications of the root was counted.

### 2.6. Statistical Analysis

Statistical data processing was performed using the software GraphPad Prism 5 and Excel of Microsoft Office 2013 version. The Kruskal–Wallis test and Dunn's multiple comparison tests were performed.

## 3. Results

### 3.1. Germination Success

Nine days after sowing, the three varieties were observed to emerge, with Amelia showing the most abundant germination. The PGPR + EM biostimulant and the two biostimulant mixtures, PGPR + *Trichoderma* + OF and PGPR + EM + OF, favored germination in Amelia or showed the same percentage of germination as the dry control (Figure 1). All the biostimulants showed a higher germination percentage than the dry control in IMIDRA10. In Blanco Sinaloa, the dry control, on the other hand, showed greater germination success than with any biostimulant. The excess humidity in the control seeds impregnated with 10 mL showed slightly lower germination percentages in IMIDRA10 and Blanco Sinaloa, but not in Amelia.

On day 32, the picture was quite different, since Blanco Sinaloa had germinated 100% in practically all treatments (Figure 2). The dry controls of the three varieties had germination percentages of 100%. In Amelia, the treatments with biostimulants also reached 100% in PGPR + *Trichoderma*, PGPR + EM, OF, and the mixtures PGPR + *Trichoderma* + OF and PGPR + EM + OF. In IMIDRA10, the treatments with biostimulants reached 100% in all cases, with the exception of PGPR + Trichoderma and the mixture of PGPR + *Trichoderma* + OF. In Blanco Sinaloa, all treatments with biostimulants reached 100% in all cases. It was found, again, that the excess humidity in the control seeds impregnated with 10 mL led to slightly lower germination percentages in IMIDRA10 and Blanco Sinaloa, but not in Amelia. On the last day of the crop, day 42, the scenario was the same in terms of germination success as on day 32.

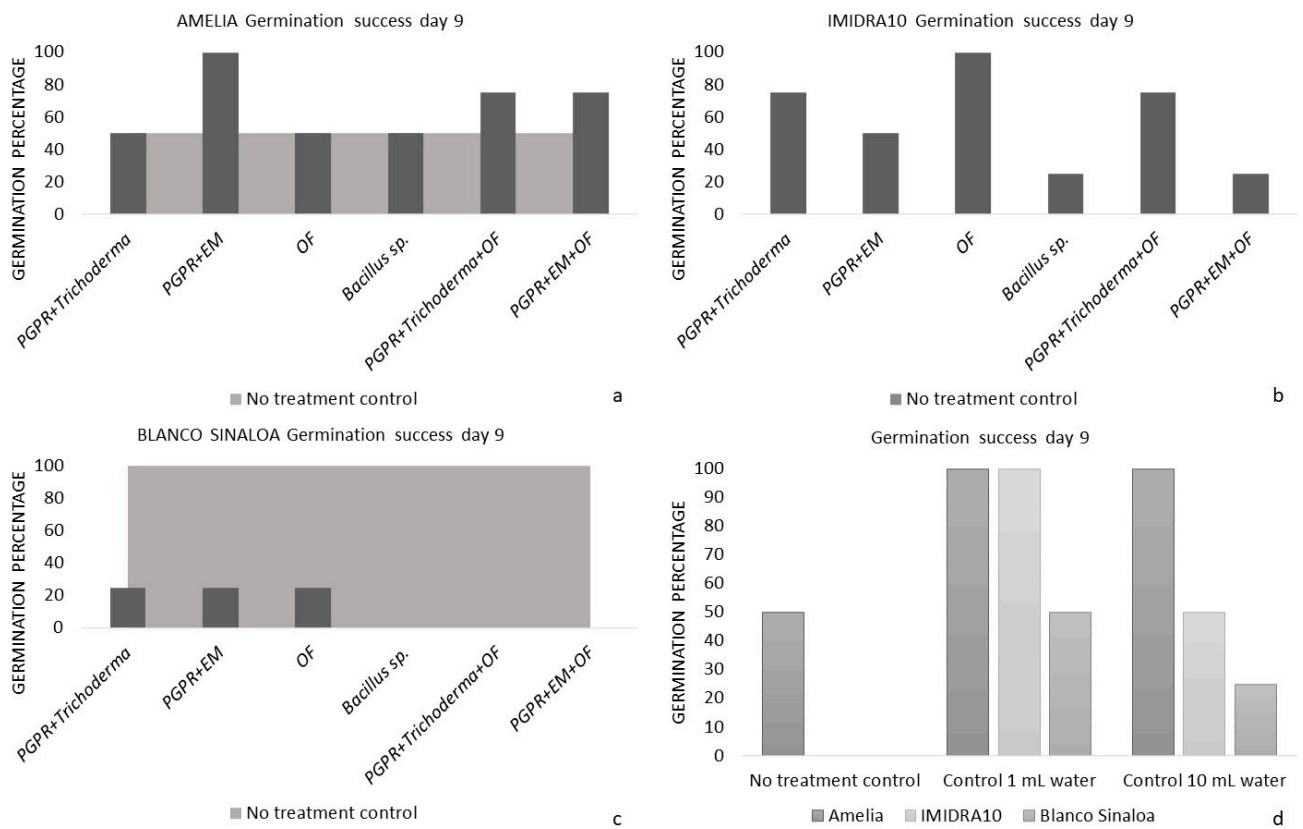

**Figure 1.** Germination success of the three chickpea varieties on day 9. (**a**) Amelia; (**b**) IMIDRA10; (**c**) Blanco Sinaloa; (**d**) comparison of no-treatment control with controls with 1 mL and 10 mL of distilled water.

### 3.2. Vegetative and Radicular Development

Regarding the vegetative development of the plants, the following results were obtained on days 32 and 42 of the culture. The results for average plant length and number of nodes on the different culture dates are shown in Tables 1 and 2, respectively. It is important to note that there were no significant differences between the results of the lengths of the seeds treated with biostimulants among themselves or with respect to the untreated control in any of the three varieties. Despite this, the following observations can be made. There were significant differences between the plant lengths among the three varieties (Kruskal–Wallis test, $p$-value = 0.0033), since Blanco Sinaloa is much larger than Amelia and IMIDRA 10. Blanco Sinaloa is a variety that requires irrigation, and it was the only one in which the plant lengths of the biostimulant-impregnated seeds were superior to those of untreated plants; that is, it was the only one that was positively affected by biostimulants, although without statistically significant differences (Kruskal–Wallis test, $p$-value = 0.4410; Figure 3). On day 32 of cultivation, Amelia and IMIDRA10 had lower plant lengths than the dry controls, the untreated chickpea seeds of the same varieties, compared to the seeds treated with different biostimulants (Kruskal–Wallis test, $p$-values = 0.5175 and 0.9254; Amelia and IMIDRA10 respectively). That is, the different treatments did not favor growth in these two varieties of plants. The untreated controls had better length results. In Amelia, the most favorable treatments were PGPR + *Trichoderma* and PGPR + EM + OF. In IMIDRA10, practically all the treatments presented results that were similar to, but lower than, those of the untreated control. In Blanco Sinaloa, the treatments also presented similar results, although the plants were higher and longer, than those of the untreated control. The most favorable treatments were PGPR + *Trichoderma* and PGPR + EM. It was observed that the excess humidity the control seeds impregnated with 10 mL showed lower vegetative development in the typical

dryland varieties, Amelia and IMIDRA10, though it was higher in the variety grown with irrigation, Blanco Sinaloa.

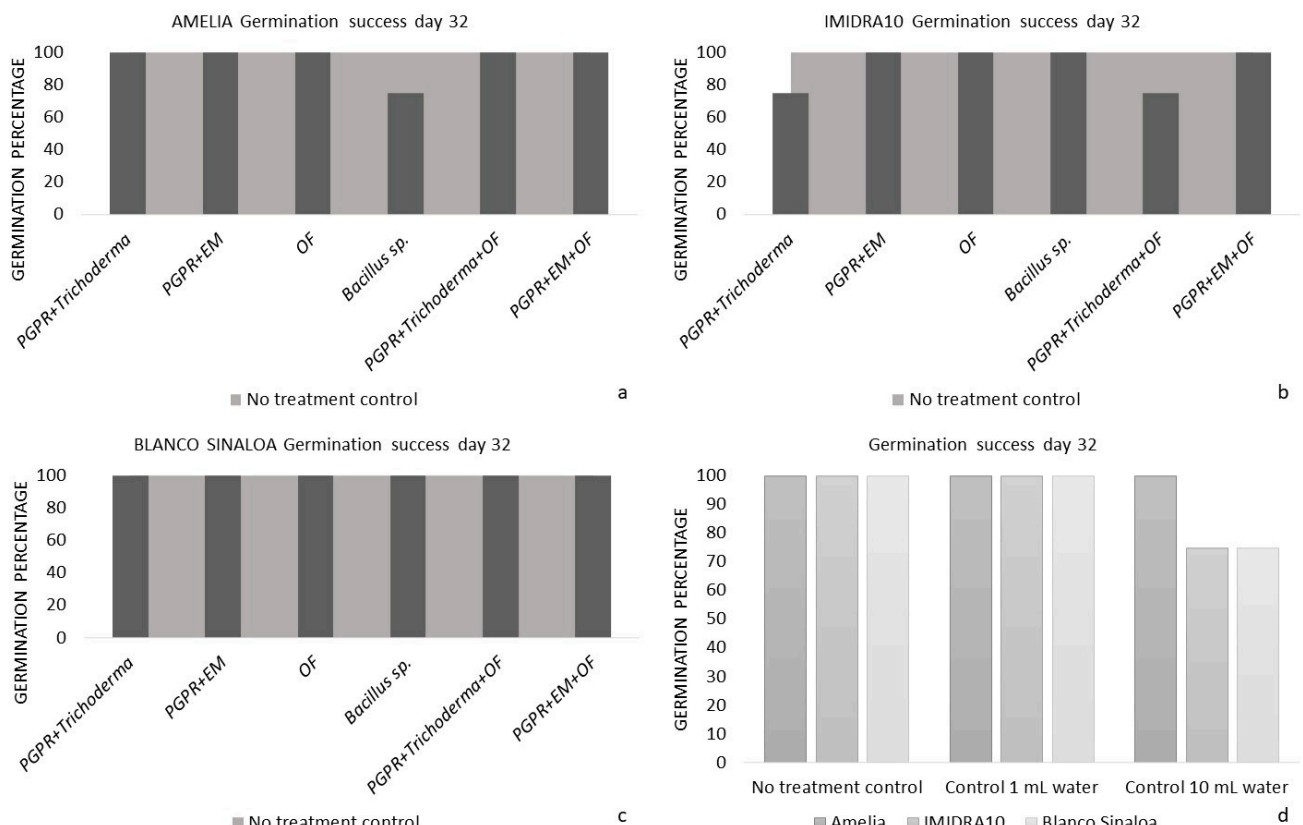

**Figure 2.** Germination success of the three chickpea varieties on day 32. (**a**) Amelia; (**b**) IMIDRA10; (**c**) Blanco Sinaloa; (**d**) comparison of no treatment control with control with 1 mL and 10 mL of distilled water.

On the last day of culture, day 42, the results were similar (Figure 4). In general, the biostimulants did not provide a great advantage in the vegetative development of the chickpeas with respect to the untreated seeds in any of the varieties. Only in Blanco Sinaloa were the lengths of the seed plants treated with the biostimulants PGPR + *Trichoderma*, PGPR + EM, and the mixture of PGPR + *Trichoderma* + OF higher than those of the untreated control (Kruskal–Wallis test, *p*-value = 0.5682). The plants whose seeds were treated with the biostimulants *Bacillus* sp., OF, and PGPR + EM + OF had similar lengths to those of the untreated seeds. In Amelia and IMIDRA10, the plants of the treated seeds had shorter lengths than those of the untreated seeds in all cases (Kruskal–Wallis test, *p*-values = 0.2849 and 0.4713; Amelia and IMIDRA10, respectively), obtaining the best results with PGPR + *Trichoderma* in Amelia and with *Bacillus* sp. in IMIDRA10. Once again, it was observed that the excess moisture in the control seeds impregnated with 10 mL showed lower vegetative development in the typical rainfed varieties, Amelia and IMIDRA10, while it was higher in the variety grown with irrigation, Blanco Sinaloa.

On day 42, the number of root ramifications was studied, and it was noteworthy that the organic fertilizer, OF, produced an increase in root ramifications with respect to the dry control in the three varieties, with the highest effect occurring in IMIDRA10 (Figure 5). As with vegetative development, it was observed that the excess humidity in the control seeds impregnated with 10 mL showed lower root development and a lower number of ramifications in the typical rainfed varieties, Amelia and IMIDRA10, while these parameters were higher in the variety grown with irrigation, Blanco Sinaloa. Nodules in the roots were not observed.

**Table 1.** Average (x̄) and standard deviation (σ) of the plant length (cm) of the different treatments on different culture dates. AM, Amelia; IM10, IMIDRA10; BS, Blanco Sinaloa.

| Dates in 2022 | 05/12 | | 07/12 | | 09/12 | | 12/12 | | 14/12 | | 16/12 | | 19/12 | | 23/12 | | 28/12 | |
|---|---|---|---|---|---|---|---|---|---|---|---|---|---|---|---|---|---|---|
| | x̄ | σ | x̄ | σ | x̄ | σ | x̄ | σ | x̄ | σ | x̄ | σ | x̄ | σ | x̄ | σ | x̄ | σ |
| No-Treatment Control AM | 5.3 | 1.6 | 7.5 | 1.7 | 8.9 | 2.0 | 10.1 | 2.2 | 12.8 | 2.8 | 14.6 | 2.9 | 16.2 | 2.9 | 17.3 | 2.6 | 20.3 | 2.7 |
| No-Treatment Control IM10 | 5.3 | 1.4 | 7.6 | 2.0 | 9.2 | 2.3 | 9.9 | 2.7 | 12.9 | 2.8 | 14.9 | 3.4 | 16.4 | 3.4 | 18.1 | 3.7 | 19.7 | 4.5 |
| No-Treatment Control BS | 3.6 | 1.8 | 6.0 | 2.4 | 8.4 | 3.3 | 10.4 | 4.2 | 13.7 | 4.5 | 16.6 | 4.7 | 19.9 | 4.2 | 24.4 | 3.3 | 28.8 | 2.6 |
| Control 1 mL AM | 5.6 | 1.0 | 7.6 | 0.7 | 9.1 | 0.6 | 10.4 | 0.6 | 13.0 | 0.2 | 14.6 | 0.5 | 16.0 | 0.5 | 17.5 | 1.1 | 20.0 | 1.4 |
| Control 1 mL IM10 | 5.5 | 0.8 | 7.8 | 0.9 | 9.5 | 1.0 | 10.5 | 1.8 | 13.4 | 1.8 | 14.5 | 2.2 | 16.5 | 2.6 | 17.2 | 2.2 | 19.4 | 3.5 |
| Control 1 mL BS | 4.3 | 0.6 | 7.1 | 1.0 | 9.3 | 1.0 | 11.5 | 1.4 | 15.9 | 1.5 | 18.2 | 1.5 | 20.9 | 1.6 | 25.4 | 2.5 | 28.1 | 2.3 |
| Control 10 mL AM | 5.3 | 0.9 | 7.3 | 0.9 | 8.8 | 0.8 | 9.5 | 1.0 | 12.1 | 1.4 | 13.0 | 1.6 | 14.8 | 1.4 | 15.8 | 1.9 | 18.2 | 2.1 |
| Control 10 mL IM10 | 4.9 | 2.5 | 7.1 | 0.3 | 8.7 | 0.6 | 9.4 | 1.0 | 11.9 | 1.0 | 12.8 | 0.9 | 14.3 | 7.2 | 15.4 | 0.6 | 16.6 | 1.2 |
| Control 10 mL BS | 5.3 | 2.8 | 8.0 | 1.0 | 10.5 | 1.2 | 12.5 | 2.0 | 16.8 | 1.0 | 18.9 | 1.9 | 22.1 | 11.1 | 26.7 | 1.5 | 30.8 | 2.5 |
| PGPR + *Trichoderma* AM | 5.1 | 0.5 | 7.4 | 0.8 | 9.1 | 1.0 | 10.2 | 1.2 | 12.6 | 1.8 | 14.2 | 2.1 | 16.3 | 2.1 | 18.1 | 2.6 | 20.0 | 2.5 |
| PGPR + *Trichoderma* IM10 | 6.3 | 3.3 | 8.2 | 1.5 | 9.6 | 1.2 | 10.1 | 1.2 | 12.6 | 1.3 | 13.7 | 1.1 | 15.3 | 7.7 | 15.7 | 1.1 | 16.6 | 1.9 |
| PGPR + *Trichoderma* BS | 5.5 | 0.6 | 8.3 | 0.3 | 10.3 | 0.3 | 13.1 | 1.0 | 18.2 | 1.0 | 20.4 | 1.0 | 22.2 | 1.0 | 27.3 | 2.4 | 30.7 | 2.5 |
| PGPR + EM AM | 4.7 | 1.3 | 6.4 | 2.2 | 7.3 | 2.4 | 7.9 | 2.9 | 9.9 | 4.3 | 10.5 | 4.9 | 11.6 | 5.7 | 15.1 | 1.9 | 17.0 | 1.3 |
| PGPR + EM IM10 | 5.8 | 0.8 | 7.7 | 0.8 | 9.2 | 1.2 | 9.9 | 1.0 | 12.8 | 1.7 | 13.9 | 1.3 | 14.8 | 1.7 | 16.2 | 2.5 | 17.3 | 3.0 |
| PGPR + EM BS | 4.7 | 0.8 | 7.4 | 1.3 | 10.0 | 1.8 | 12.8 | 1.2 | 16.7 | 1.4 | 19.2 | 1.6 | 22.1 | 1.1 | 25.8 | 1.4 | 29.9 | 1.5 |
| OF AM | 5.2 | 1.0 | 7.0 | 1.0 | 8.7 | 1.3 | 9.5 | 1.6 | 11.5 | 1.0 | 13.4 | 1.2 | 15.0 | 1.5 | 16.4 | 1.0 | 18.4 | 0.3 |
| OF IM10 | 5.8 | 0.9 | 7.5 | 0.9 | 9.4 | 0.8 | 9.6 | 0.8 | 12.1 | 0.4 | 13.1 | 0.7 | 14.7 | 0.4 | 15.4 | 1.0 | 16.8 | 1.8 |
| OF BS | 4.7 | 0.5 | 6.9 | 0.5 | 9.1 | 0.6 | 11.6 | 0.9 | 15.9 | 1.1 | 17.9 | 1.4 | 20.8 | 0.9 | 24.1 | 2.0 | 26.6 | 3.2 |
| *Bacillus* AM | 5.2 | 2.6 | 6.9 | 0.8 | 8.6 | 0.6 | 9.2 | 1.1 | 11.2 | 1.3 | 12.1 | 1.2 | 13.5 | 6.9 | 14.4 | 2.1 | 16.1 | 2.5 |
| *Bacillus* IM10 | 5.1 | 0.4 | 7.1 | 0.3 | 9.0 | 0.5 | 9.4 | 0.5 | 12.5 | 1.0 | 13.7 | 1.4 | 15.5 | 1.1 | 17.2 | 2.0 | 18.7 | 2.5 |
| *Bacillus* BS | 5.4 | 1.7 | 8.0 | 2.7 | 10.1 | 2.7 | 12.7 | 2.9 | 16.8 | 3.6 | 18.6 | 3.6 | 21.2 | 2.8 | 25.7 | 3.1 | 28.6 | 2.7 |
| PGPR + *Trichoderma* + OF AM | 5.6 | 0.9 | 7.8 | 1.1 | 8.9 | 1.2 | 10.3 | 0.7 | 11.9 | 0.8 | 12.9 | 1.1 | 14.2 | 1.5 | 15.6 | 1.9 | 17.5 | 2.9 |
| PGPR + *Trichoderma* + OF IM10 | 5.9 | 3.1 | 8.2 | 1.2 | 9.5 | 0.8 | 10.9 | 1.0 | 13.1 | 1.0 | 13.5 | 1.5 | 14.7 | 7.4 | 15.4 | 0.5 | 16.5 | 0.2 |
| PGPR + *Trichoderma* + OF BS | 4.7 | 1.4 | 7.1 | 1.6 | 9.3 | 1.9 | 11.6 | 2.0 | 16.0 | 2.6 | 18.0 | 2.7 | 20.9 | 2.5 | 26.2 | 1.7 | 29.6 | 1.2 |
| PGPR + EM + OF AM | 6.5 | 1.0 | 8.9 | 1.0 | 10.4 | 1.4 | 11.3 | 1.6 | 13.2 | 1.6 | 14.4 | 1.5 | 15.7 | 1.8 | 17.2 | 1.7 | 19.0 | 2.4 |
| PGPR + EM + OF IM10 | 5.6 | 0.5 | 7.6 | 0.7 | 9.1 | 0.6 | 10.1 | 0.9 | 12.4 | 0.7 | 13.1 | 0.7 | 14.1 | 1.0 | 15.1 | 0.3 | 16.5 | 0.2 |
| PGPR + EM + OF BS | 5.2 | 1.7 | 7.9 | 2.2 | 10.0 | 2.9 | 12.2 | 3.9 | 15.9 | 4.2 | 17.5 | 4.2 | 20.4 | 3.4 | 24.7 | 2.7 | 28.2 | 2.9 |

**Table 2.** Average (x̄) and standard deviation (σ) of the number of nodes of the different treatments on different culture dates. AM, Amelia; IM10, IMIDRA10; BS, Blanco Sinaloa.

| Dates in 2022 | 29/11 | | 02/12 | | 05/12 | | 07/12 | | 09/12 | | 12/12 | | 16/12 | | 21/12 | | 28/12 | |
|---|---|---|---|---|---|---|---|---|---|---|---|---|---|---|---|---|---|---|
| | x̄ | σ | x̄ | σ | x̄ | σ | x̄ | σ | x̄ | σ | x̄ | σ | x̄ | σ | x̄ | σ | x̄ | σ |
| No-Treatment Control AM | 2.3 | 1.0 | 3.3 | 1.0 | 4.3 | 1.0 | 4.5 | 0.6 | 5.3 | 1.0 | 6.5 | 0.6 | 6.8 | 0.5 | 7.8 | 0.5 | 9.5 | 1.0 |
| No-Treatment Control IM10 | 1.5 | 0.6 | 2.8 | 1.0 | 3.5 | 0.6 | 4.3 | 0.5 | 5.3 | 0.5 | 5.8 | 0.5 | 6.3 | 0.5 | 7.3 | 0.5 | 9.5 | 0.6 |
| No-Treatment Control BS | 1.3 | 0.5 | 2.3 | 1.0 | 3.3 | 1.0 | 3.8 | 1.3 | 4.8 | 1.3 | 6.3 | 1.0 | 6.8 | 0.5 | 7.8 | 0.5 | 10.0 | 0.8 |
| Control 1 mL AM | 1.8 | 0.5 | 3.3 | 0.5 | 3.8 | 0.5 | 4.5 | 0.6 | 5.3 | 0.5 | 5.5 | 0.6 | 6.5 | 0.6 | 7.5 | 0.6 | 9.8 | 1.7 |
| Control 1 mL IM10 | 2.3 | 1.0 | 3.8 | 0.5 | 3.8 | 0.5 | 4.3 | 0.5 | 5.5 | 0.6 | 6.5 | 0.6 | 6.5 | 0.6 | 7.3 | 0.5 | 9.3 | 0.5 |
| Control 1 mL BS | 1.3 | 0.5 | 2.5 | 1.0 | 3.8 | 0.5 | 4.0 | 0.0 | 4.8 | 0.5 | 5.5 | 0.6 | 6.8 | 0.5 | 8.0 | 0.0 | 10.0 | 0.8 |
| Control 10 mL AM | 2.3 | 0.5 | 3.3 | 0.5 | 4.0 | 0.0 | 4.8 | 0.5 | 5.5 | 0.6 | 6.5 | 0.6 | 6.3 | 0.5 | 7.0 | 0.0 | 9.5 | 1.0 |
| Control 10 mL IM10 | 1.0 | 0.8 | 2.3 | 0.0 | 3.0 | 2.0 | 3.5 | 0.6 | 6.0 | 0.0 | 6.0 | 0.0 | 4.5 | 3.0 | 5.3 | 3.5 | 9.0 | 1.0 |
| Control 10 mL BS | 1.5 | 1.3 | 2.3 | 0.0 | 2.8 | 1.9 | 3.3 | 0.6 | 5.7 | 0.6 | 6.7 | 0.6 | 5.0 | 3.4 | 5.8 | 3.9 | 10.3 | 0.6 |
| PGPR + *Trichoderma* AM | 1.5 | 0.6 | 3.0 | 0.8 | 3.8 | 0.5 | 4.3 | 0.5 | 5.5 | 0.6 | 5.8 | 0.5 | 6.3 | 0.5 | 7.0 | 0.0 | 9.3 | 0.5 |
| PGPR + *Trichoderma* IM10 | 2.3 | 1.7 | 2.8 | 0.6 | 3.3 | 2.2 | 3.5 | 0.6 | 6.3 | 0.6 | 6.7 | 0.6 | 4.8 | 3.2 | 5.3 | 3.5 | 9.7 | 0.6 |
| PGPR + *Trichoderma* BS | 2.0 | 0.0 | 3.0 | 0.0 | 3.5 | 0.6 | 4.8 | 0.5 | 6.0 | 0.8 | 6.3 | 1.0 | 6.8 | 0.5 | 7.8 | 0.5 | 10.3 | 1.3 |
| PGPR + EM AM | 2.3 | 1.0 | 3.5 | 0.6 | 3.8 | 0.5 | 4.3 | 1.0 | 5.5 | 1.7 | 6.8 | 0.5 | 6.3 | 1.5 | 5.8 | 3.9 | 9.7 | 1.2 |
| PGPR + EM IM10 | 2.3 | 1.0 | 3.5 | 0.6 | 4.0 | 0.0 | 4.3 | 0.5 | 6.3 | 0.5 | 6.5 | 0.6 | 6.3 | 0.5 | 7.3 | 0.5 | 9.0 | 0.8 |
| PGPR + EM BS | 1.5 | 1.0 | 3.0 | 0.8 | 3.3 | 0.5 | 4.3 | 0.5 | 6.3 | 0.5 | 6.8 | 0.5 | 7.0 | 0.0 | 8.0 | 0.0 | 10.5 | 1.0 |
| OF AM | 1.8 | 1.0 | 3.0 | 0.0 | 3.8 | 0.5 | 4.8 | 0.5 | 5.5 | 0.6 | 5.8 | 0.5 | 6.3 | 0.5 | 7.0 | 0.0 | 9.3 | 0.5 |
| OF IM10 | 3.0 | 0.0 | 3.8 | 0.5 | 4.0 | 0.0 | 4.8 | 0.5 | 6.0 | 0.0 | 6.8 | 0.5 | 6.3 | 0.5 | 7.0 | 0.0 | 9.3 | 1.0 |
| OF BS | 1.5 | 0.6 | 2.3 | 0.5 | 3.3 | 0.5 | 4.0 | 0.0 | 5.3 | 0.5 | 6.0 | 0.0 | 6.8 | 0.5 | 7.5 | 0.6 | 9.8 | 1.0 |
| *Bacillus* AM | 1.5 | 1.0 | 2.3 | 0.0 | 3.0 | 2.0 | 3.8 | 0.0 | 5.3 | 0.6 | 7.0 | 0.0 | 4.8 | 3.2 | 5.5 | 3.7 | 9.7 | 1.2 |
| *Bacillus* IM10 | 2.3 | 1.0 | 3.3 | 0.5 | 3.5 | 0.6 | 4.0 | 0.0 | 5.8 | 0.5 | 6.0 | 0.0 | 6.3 | 0.5 | 7.0 | 0.0 | 9.0 | 0.8 |

**Table 2.** *Cont.*

| Dates in 2022 | 29/11 | | 02/12 | | 05/12 | | 07/12 | | 09/12 | | 12/12 | | 16/12 | | 21/12 | | 28/12 | |
|---|---|---|---|---|---|---|---|---|---|---|---|---|---|---|---|---|---|---|
| | x̄ | σ | x̄ | σ | x̄ | σ | x̄ | σ | x̄ | σ | x̄ | σ | x̄ | σ | x̄ | σ | x̄ | σ |
| *Bacillus* BS | 1.8 | 1.0 | 3.0 | 0.8 | 3.8 | 0.5 | 4.5 | 0.6 | 6.0 | 0.0 | 6.5 | 0.6 | 7.3 | 0.5 | 8.0 | 0.0 | 10.0 | 0.0 |
| PGPR + *Trichoderma* + OF AM | 2.3 | 1.0 | 3.3 | 1.0 | 3.8 | 0.5 | 4.5 | 0.6 | 5.5 | 0.6 | 6.0 | 0.8 | 6.3 | 0.5 | 7.0 | 0.0 | 9.0 | 0.8 |
| PGPR + *Trichoderma* + OF IM10 | 2.0 | 1.4 | 2.5 | 0.6 | 3.0 | 2.0 | 3.3 | 0.6 | 6.7 | 0.6 | 7.0 | 0.0 | 5.0 | 3.4 | 5.3 | 3.5 | 9.7 | 0.6 |
| PGPR + *Trichoderma* + OF BS | 1.3 | 1.0 | 2.3 | 1.0 | 3.0 | 0.8 | 3.8 | 0.5 | 5.3 | 0.5 | 5.3 | 0.5 | 6.8 | 0.5 | 7.3 | 0.5 | 10.0 | 0.0 |
| PGPR + EM + OF AM | 3.0 | 0.0 | 3.8 | 0.5 | 4.5 | 0.6 | 4.8 | 0.5 | 5.8 | 0.5 | 6.5 | 0.6 | 6.8 | 0.5 | 7.5 | 0.6 | 9.5 | 0.6 |
| PGPR + EM + OF IM10 | 2.0 | 0.0 | 3.0 | 0.0 | 4.0 | 0.0 | 4.3 | 0.5 | 5.8 | 1.0 | 6.0 | 0.8 | 6.3 | 0.5 | 7.0 | 0.0 | 8.5 | 0.6 |
| PGPR + EM + OF BS | 1.8 | 0.5 | 2.8 | 0.5 | 3.8 | 0.5 | 4.3 | 1.0 | 5.8 | 0.5 | 6.8 | 0.5 | 6.8 | 0.5 | 7.8 | 0.5 | 9.8 | 0.5 |

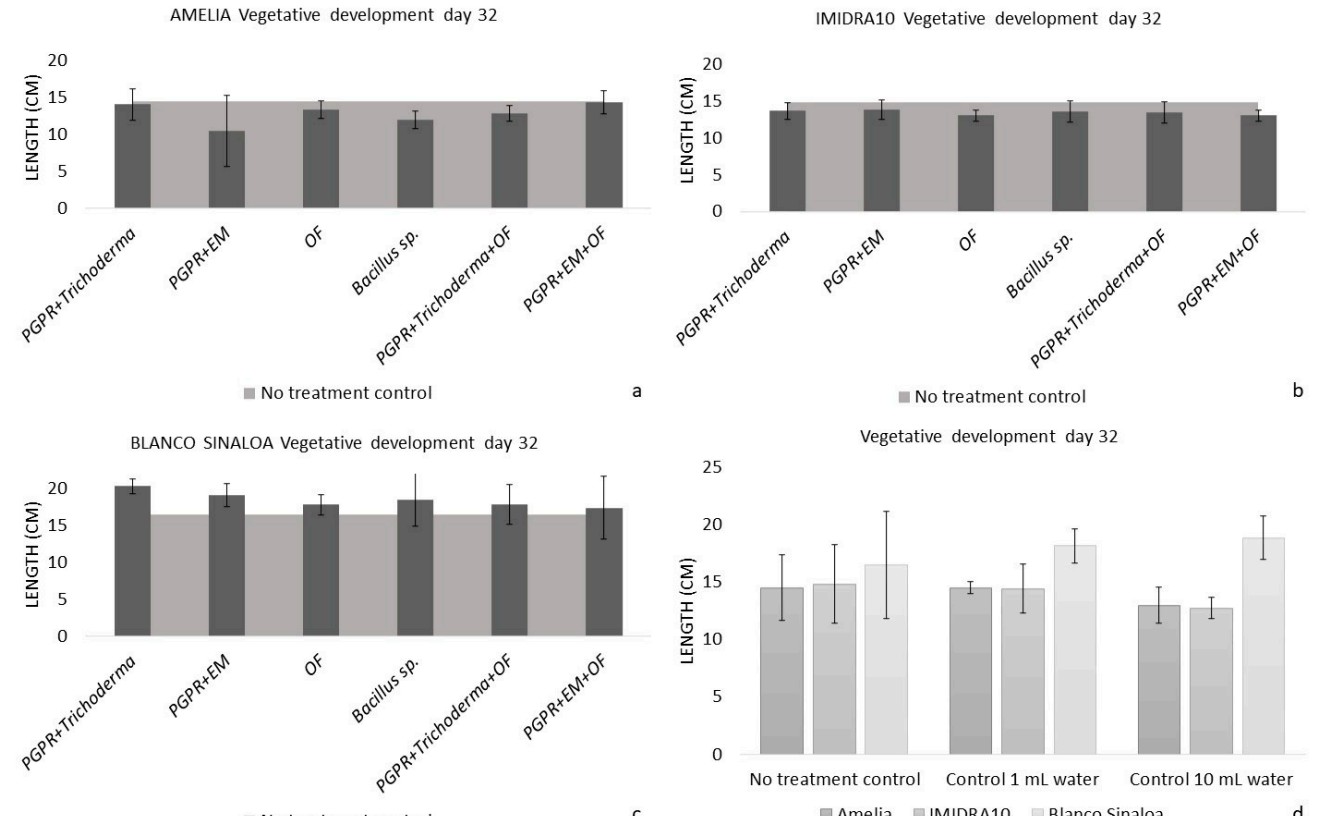

**Figure 3.** Vegetative development of the three chickpea varieties on day 32. (**a**) Amelia; (**b**) IMIDRA10; (**c**) Blanco Sinaloa; (**d**) comparison of no-treatment control with control with 1 mL and 10 mL of distilled water.

It was observed that the lengths of the plants of Blanco Sinaloa were much greater than those of Amelia and IMIDRA10. In general, the longer lengths were obtained with PGPR + *Trichoderma* treatment impregnated in the seeds, followed by PGPR + EM and, finally, the untreated control (Figure 6).

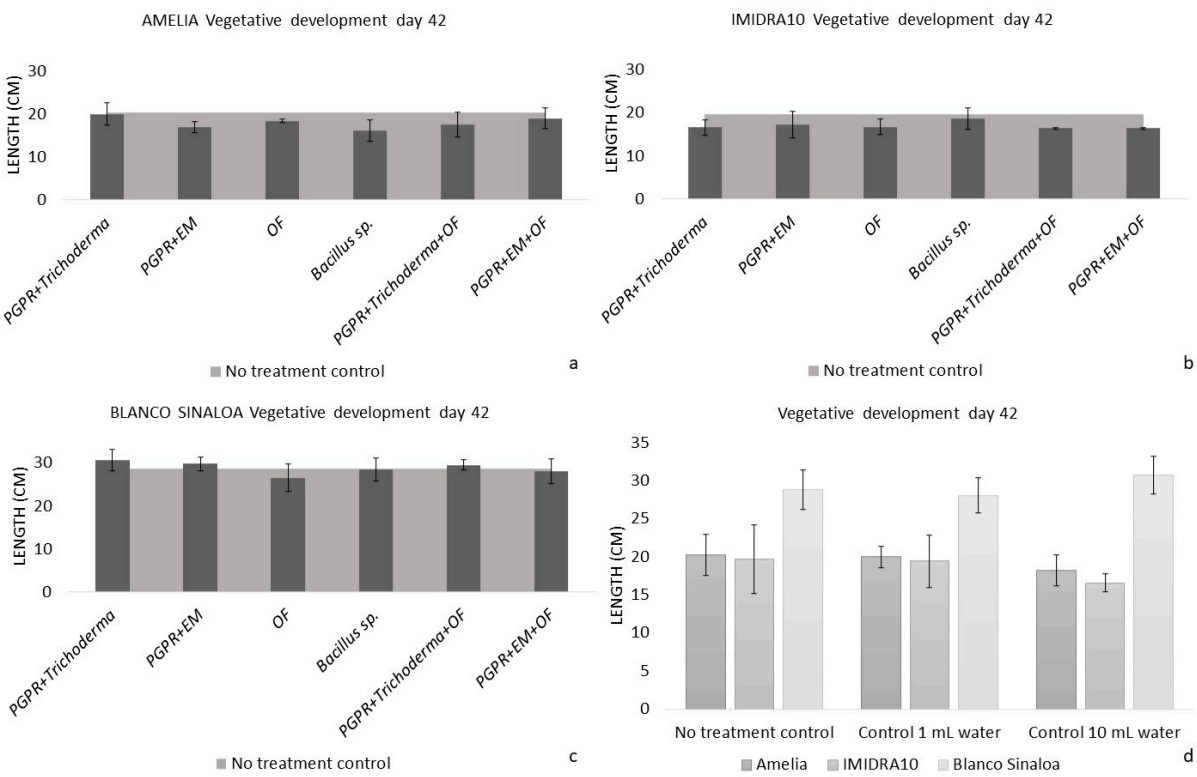

**Figure 4.** Vegetative development of the three chickpea varieties on day 42. (**a**) Amelia; (**b**) IMIDRA10; (**c**) Blanco Sinaloa; (**d**) comparison of no-treatment control with controls with 1 mL and 10 mL of distilled water.

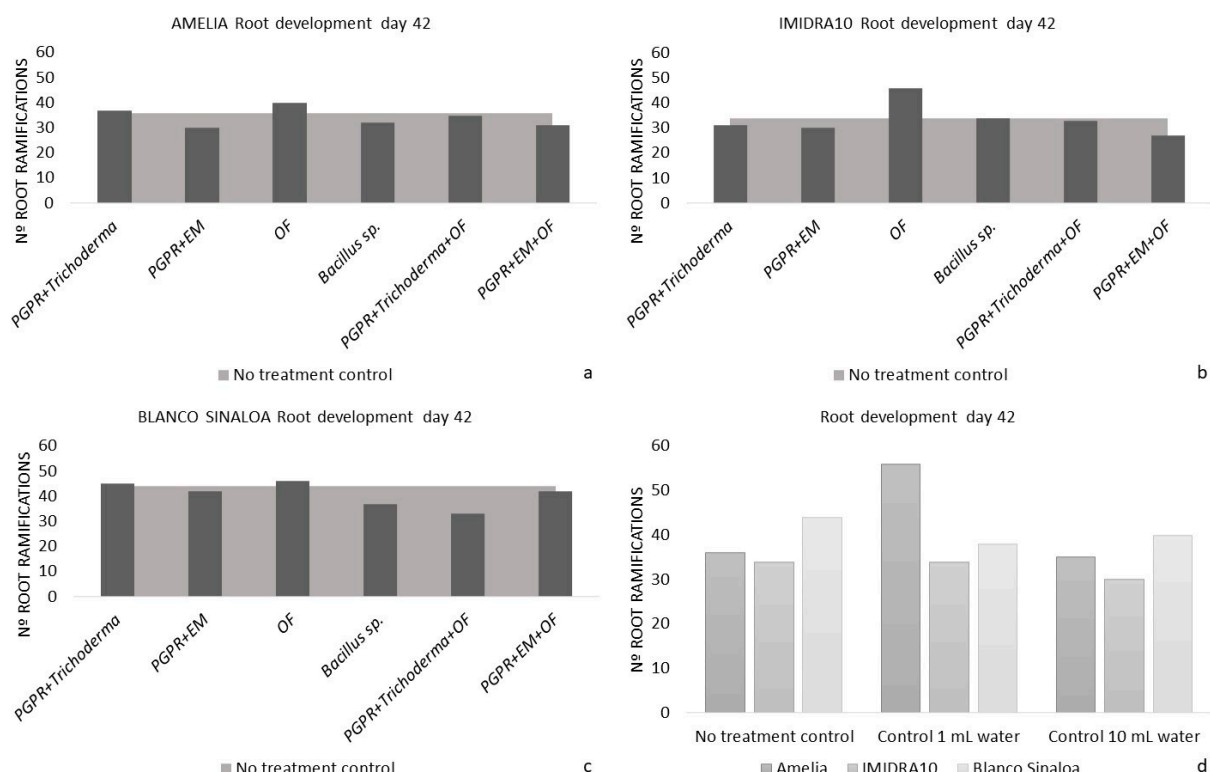

**Figure 5.** Radicular development and number of root ramifications of the three chickpea varieties on day 42. (**a**) Amelia; (**b**) IMIDRA10; (**c**) Blanco Sinaloa; (**d**) comparison of no-treatment control with controls with 1 mL and 10 mL of distilled water.

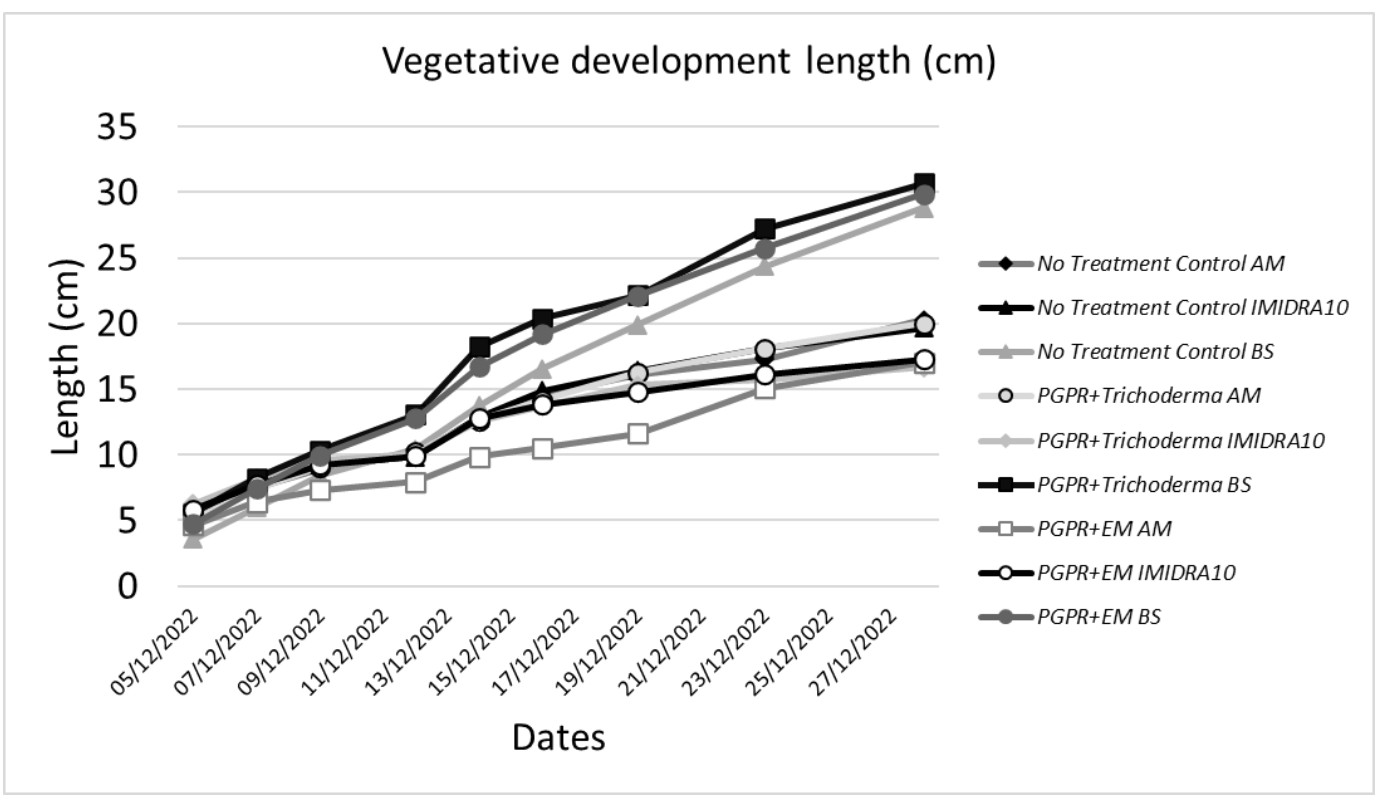

**Figure 6.** Vegetative development of the plants with different treatments.

## 4. Discussion

Different effects of the biostimulants on the distinct varieties of chickpeas which we analyzed were observed. The positive effect of the biostimulants was prominent in the chickpea variety used for irrigation, Blanco Sinaloa. The length of the plants and number of nodes were higher when the seeds were impregnated with biostimulants compared with non-treated seeds. The two chickpea varieties cultured under rainfed conditions, Amelia and IMIDRA10, did not show such influence by biostimulants. The length of the plants and the number of nodes from seeds impregnated with biostimulants were, in all cases, inferior to non-treated seeds. It seems as if the variety grown with irrigation is more sensitive to the plant growth hormones and vitamins provided by the biostimulants. Most of the biostimulants used in this experiment were composed of PGPR (*Bacillus* sp. is another PGPR), which is involved in the production and release of hormone molecules that stimulate plant health and growth, such as giberelines and cytokines [3,26,27]. This is likely a very important effect of biostimulants based on PGPR. Moreover, they fix nitrogen from the atmosphere and increase the bioavailability of phosphorus for plants [27]. Only the organic fertilizer, OF, has no PGPR, as it is a nitrogen and potassium (NK) liquid organic fertilizer of plant origin with a high content of organic matter. The composition of the organic matter is not specified, but it could also contain some plant hormones or vitamins. Some research has been undertaken to understand the molecular pathways and alterations in the expression of genes in the plants at the molecular level by different biostimulants [26], but further research must be carried out. There are studies on the effect of *Bacillus* spp. on plants [28–30] reporting these bacteria acting as biofungicides, promoting plant and soil health [28–30], solubilizing essential nutrients to simpler forms for root uptake [31], and producing growth-promoting substances such as cytokinins, spermidines, gibberellins, and IAA [31]. As a curiosity, different strains of *Bacillus* spp. are able to regulate nitrogen concentrations in soil. Some strains fix it from the atmosphere [32,33], while other strains mitigate the negative effects of high N concentrations in the soil for roots because of the use of N by the bacteria itself [34,35]. *B. amyloquefaciens* upregulates the NHX1, NHX7,

$H^+$-PPase, and HKT1 genes, which indicates that it plays an active role in the sequestration of $Na^+$ [36]. *Pseudomonas fluorescens* strain LBUM677 increased seed weight and number, as well as the oil content in *Brassica napus, Buglossoides arvensis,* and *Glycine max*, because it was attributed to produce ACC deaminase and IAA and solubilize micronutrients [37]. *P. fluorescens* also increased Ca, Mg, K, P, and Zn concentrations in *Amaranthus hybridus* L. leaves [38]. *Rhizobium* spp. is also known for producing secondary metabolites and plant growth hormones [39], aside from reducing atmospheric nitrogen and solubilizing nutrients. The inoculation of chickpea plants with *Rhizobium* sp. combined with foliar application of $GA_3$ significantly enhanced plant biomass and yield up to 39% [40], and also increased the chlorophyll and NPK content of the plants and the nutritional content of chickpea seeds. *Rhizobium* sp. synthesizes hormones such as gibberellins and IAA [41]. *Rhizobium* spp. could also have positive effects on non-legume species of plants [42].

The fungi *Trichoderma*, present in two of the biostimulants used in this essay, stimulates the growth of the radicular system and improves the immune system of the plant. Several *Trichoderma* spp. strains improve tolerance to abiotic stresses and increase plant growth, development, and yield [11–13]. *Trichoderma virens* GV41-based biostimulants have been shown to increase phenol content, antioxidant activity, and nitrogen usage efficiency in lettuce [43]. Curiously, the biostimulant that promoted a greater number of root ramifications in the present assay was the organic fertilizer, OF, which did not contain *Trichoderma* or the fungus *Glomus intraradices, which* forms endomycorrhizas [14]. *Glomus intraradices* was present in two of the other biostimulants used. We did not observe nodules in any roots, which is strange for chickpea plants. This could be due to the substratum used for the assay. Maybe it was very rich in nutrients, and roots did not need to form nodules. The formation of mycorrhizas with different biostimulants will studied in the next experiment with different types of soil to attempt to answer this question. The molecular mechanisms of the positive effects of PGPB and *Trichoderma* spp. as biostimulants remain undiscovered [26]. The application of *Glomus intraradices* in wheat crops increased the plant height in the greenhouse, but did not have any effect on the characteristics of the plants, mycorrhizal colonization, yield, or grain quality in the field [14]. Treatments with *Glomus intraradices* in maize had no effect with respect to the negative control [44].

According to past results [45], chickpea grains should be treated with biostimulants in such a way that seeds are dry for the sowing, because the mechanic seeder only works with dry seeds. Because of this, the biostimulant was applied, mixed with water, to the seeds 24 h before sowing, and then the seeds were left to dry. This biostimulant impregnation was different to seed priming, because the seeds were not soaked in the biostimulant for 24 h [46,47]. In the present work, 100 g of seeds were impregnated with just 1 mL of the different types of biostimulants and then left to dry for 24 h. The use of biostimulants as seed priming agents should be investigated in the future. Some commercial trades suggest applying the biostimulant onto the seeds mixed with some water, which is considered the best method if a mechanic seeder is used afterward, around 24 h previous to the sowing, as was previously mentioned. Another suggested way to apply the biostimulant is mixed with water in irrigation, which is easy if the culture is irrigated, but increases the cost of sowing if the culture is under rainfed conditions. Another disadvantage of this method of applying the biostimulant is the probable promotion of weeds in the field [48–50]. It is important to find a balance of enriching the soil to increase the vitality of the chickpea culture with increasing the growth of other adventive plants that compete with the chickpea. It is the same disjunctive as applying fertilizer to the soil which, in the case of chickpeas and other legumes, was mycorrhized naturally by *Rhizobium* sp. Sometimes there are no differences in production between fertilized and unfertilized plots [5], and fertilizer also promotes weeds that compete with the chickpea plants. Weeds have been shown to be more competitive with chickpea than with other crops such as canola, fababean, or wheat [51]. The same disadvantage of promoting weeds is shared with other suggested methods of application of the biostimulant, which is in the form of powder mixed with soil. It could also promote weeds along with increasing the vitality of chickpea plants.

## 5. Conclusions

Blanco Sinaloa is a chickpea variety that requires irrigation, and it is the only one in which the plant lengths of the biostimulant-impregnated seeds were superior to those of the untreated plants, although without statistically significant differences. The Amelia and IMIDRA10 varieties, when treated with the different biostimulants, had lower plant lengths than the untreated chickpea seeds. That is, the different treatments did not favor growth in either of these two rainfed varieties. It seems as if the variety used for irrigation, Blanco Sinaloa, is more sensitive to the biostimulants. The mechanic seeder only works with dry seeds, so biostimulants were impregnated 24 h before sowing such that seeds would be dry for the sowing.

**Author Contributions:** This work was carried out in collaboration with all authors. Conceptualization, R.G., I.M.-M., P.M. and J.S.; methodology, R.G., I.M.-M., P.M. and J.S.; formal analysis, R.G. and I.M.-M.; resources, R.G. and I.M.-M.; writing—original draft preparation, R.G.; writing—review and editing, R.G., I.M.-M., P.V.M., D.M.-C. and J.M.; supervision, P.V.M. and J.M.; project administration, P.V.M. and J.M.; funding acquisition, P.V.M. and J.M. All authors have read and agreed to the published version of the manuscript.

**Funding:** The organization responsible for this content is GO TecnoGAR. This research has been funded under the project GO-TecnoGAR, 80% co-financed by the EU (European Agricultural Fund for Rural Development, EAFRD), and 20% co-financed by the Spanish Ministry of Agriculture, Fisheries and Food. The Directorate General for Rural Development, Innovation and Agro-Food Training (DGDRIFA) is the management authority in charge of the application of the corresponding EAFRD and national aid. Total financing €432,329.05. Jorge Sánchez Hernández is a recipient of a training fellowship funded by IMIDRA.

**Institutional Review Board Statement:** Not applicable because this study did not involve humans or animals.

**Data Availability Statement:** The average data presented in this study are contained within the article and presented in the tables and graphs. Data supporting reported results are available on request from the corresponding author.

**Acknowledgments:** We want to acknowledge all the administrative and technical support to IMIDRA personnel.

**Conflicts of Interest:** The authors declare no conflict of interest.

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
