# Peer review of "Evaluation of Biostimulatory Activity of Commercial Formulations on Three Varieties of Chickpea"

_agriculture, doi:10.3390/agriculture13020474_

Round 1

Reviewer 1 Report

Dear Authors,

The manuscript entitled: "Different effects of biostimulants depending on the chickpea variety" describes the effect of pre-sowing stimulation of chickpea seeds on germination, vegetative, and radicular development. Here are my comments on the manuscript:

- the figures are illegible, the colors are too bright, and the inscriptions are not visible, I recommend their thorough improvement

- Introduction:

Lines 30-31 with the statement "Our group is...." should not start the Introduction chapter. I would suggest starting this chapter with general information about biostimulants, or chickpea. There should be no text in the Introduction (see lines 65-87 - this should be in Materials and Methods). There is a lack of a separate research hypothesis and the purpose of the work

- Materials and Methods - please divide this section into individual subsections. Please provide information on the conditions under which the seed germination test was carried out

- I would suggest in the case of Lines 103, 109, 114, 118, and 121 "In the article it is mentioned..." not to use this term, instead I propose to write the abbreviation in brackets after the full name

- The discussion and conclusions are correct

Author Response

Responses to referee 1:

  1. Figures have been improved.
  2. Text lines 30-31 has been changed.

“Our group…” sentence has been substituted by: To improve the efficiency of chickpea (Cicer arietinum L.) crop and adapt it to climate change, the effect of biostimulants on germination success and vegetative development has been studied.

  1. Lines 65-87 have been moved to Material and Methods from Introduction.
  2. A separate research hypothesis has been clearly stated: The proposed research hypothesis is the existence of a positive influence of the different biostimulants on chickpea in terms of germination and vegetative development. The null hypothesis is the absence of biostimulant influence on chickpea germination and vegetative development.

Separated from the purpose of work: The purpose of the present work is the study of the germination success (percentage and date after the sowing) and vegetative development (plant length and number of nodes) when different biostimulants were applied in liquid form on the seeds 24 h before sowing on three different varieties of chickpea: Amelia, IMIDRA10, recently developed, and Blanco Sinaloa. Another purpose of the research study is to contribute in the development of a protocol of biostimulant seed impregnation useful for all types of agricultural practices including intensive methods.

  1. Material and Methods were divided into individual subsections.
  2. Conditions under which the seed germination test was carried out were the same as the ones described for the greenhouse assay to analyse the vegetative development of chickpea plants and are described in Material and Methods in the subsection “Greenhouse assay”.
  3. Abbreviations of the different types of biostimulants were written before its description in Material and Methods subsection 2.2. Biostimulant formulations.

Reviewer 2 Report

The manuscript entitled “Different effects of biostimulants depending on the chickpea variety”, authored by Regina Gabilondo and colleagues, deals with the investigation of the potential effect derived from the application of several biostimulant topologies on different chickpea varieties. The authors describe a variety-dependent effect, to date little described and not well known. Consequently, I believe that the manuscript has great potential for publication in Agriculture. However, a number of changes are strongly suggested before the manuscript can be judged acceptable as publication in Agriculture.

In particular,

·         Some keywords should be changed. The utility of these terms is to facilitate the search of the article using common scientific search engines (PubMed, GoogleScholar, Scopus, etc.), which rely on the terms contained in title, abstract, and keywords. Consequently, using terms that are already in these sections as keywords is inappropriate. I strongly suggest that the repetitive keywords be changed before re-submission.

·         The title should be changed. It appears to be unclear and misleading. A potential title could be "Evaluation of biostimulatory activity of commercial formulations on 7 varieties of chickpea," or something like that.

·         the affiliations section should include each author's email, along with their acronym. This acronym should be the same one used in the contributions section.

·         The article should be written in a completely impersonal way. Phrases such as "Our group," "our data," "our findings" should be seriously replaced by the main text.

·         Pointed lists should also be avoided, and the text made more discursive.

·         In the introduction section, the authors should better clarify the different types of existing biostimulants, their nature and composition. This information is very important, since they aim to use only certain types of biostimulants. The authors may have tried to include something like this at line 44, but they abruptly truncated it without elaborating.

·         information regarding the content of biostimulants described in the label should be added at least as supplementary material. I am referring to, EC, mineral composition, etc..

·         the material and methods section should be divided into several sub-sections. For example: 2.1. plant material used for the study; 2.2. Biostimulant formulations used for testing; 2.3. germination test etc. Consequently, this section should be completely rewritten.

·         Graphics should be seriously improved, since they are not at all understandable. For example, authors could opt to use color, since the journal does not charge extra for publishing colorful images.

·         The effect evaluated by the authors on seeds is similar to that of priming. Authors in the introduction should include information regarding seed priming and refer to work in which this technique has been used. For example, (i) 10.1016/j.plaphy.2021.07.015; (ii) 10.3390/agriculture10110498; (iii) 10.1016/j.plaphy.2011.11.005

·         In the graphs and tables a statistical treatment is completely missing. Indeed, this part is not even described in the materials and methods. Did the authors use statistical methods in order to validate their data?

·         The concussion section is too summary, and does not highlight the main results obtained. It should be better articulated.

Author Response

Responses to referee 2:

  1. Keywords were changed as the following: Cicer arietinum L.; Blanco Sinaloa; Amelia; PGPR; Plant Growth-Promoting Rhizobacteria; Trichoderma; Bacillus; Glomus; endomycorrhiza.
  2. Title was changed into: Evaluation of biostimulatory activity of commercial formulations on three varieties of chickpea.
  3.  Email of each author and contribution section were included.
  4. Article is now written in a completely impersonal way.
  5. Pointed lists were rephrased into a more discursive text.
  6. Different types of biostimulants were clarified with their nature and composition in the Introduction section.
  7. Information of the biostimulants in the label was added in the Material and Methods.
  8. Material and Methods was rewritten into several subsections: Chickpea varieties investigated; Biostimulant formulations; Protocol of seed impregnation with biostimulants; Greenhouse assay; Variables studied; Statistical analysis.
  9. Graphics were improved.
  10. The biostimulant was applied mixed with water on the seeds 24 h before sowing and then let the seeds dry. This biostimulant impregnation is different to seed priming because seeds are not soaked in the biostimulant for 24 h. In the present work, 100g of seeds were impregnated with just only 1ml of the different types of biostimulants and then let them dry for 24 h. The use of biostimulants as seed priming agents would be investigated in the future. Two of the references suggested about seed priming were added to the Discussion.
  11. Statistical treatment was included in graphs in vegetative development (SD) and both tables (SD). Statistical data processing was performed using the software GraphPad Prism 5 and Kruskal-Wallis test and Dunn’s Multiple Comparison Test were performed.
  12. Conclusion is now better articulated: it now highlight the results obtained.

Round 2

Reviewer 1 Report

Dear Authors,

The manuscript has been significantly bound as suggested. The figures are also of better quality. My only minor remark concerns the font on the vertical axis. Please change the font size on this axis as it is still invisible.

Author Response

Responses to referee 1:

  1. The figures were changed and now the font size of the vertical axis has been increased.

Reviewer 2 Report

The authors modified the manuscript according to the suggestions. It is now acceptable as a publication.

Author Response

Thanks.